# Influence of Isothermal Annealing on Microstructure, Morphology and Oxidation Behavior of AlTiSiN/TiSiN Nanocomposite Coatings

**DOI:** 10.3390/nano13030474

**Published:** 2023-01-24

**Authors:** Patrik Šulhánek, Libor Ďuriška, Marián Palcut, Paulína Babincová, Martin Sahul, Ľubomír Čaplovič, Martin Kusý, Ľubomír Orovčík, Štefan Nagy, Leonid Satrapinskyy, Marián Haršáni, Ivona Černičková

**Affiliations:** 1Faculty of Materials Science and Technology in Trnava, Slovak University of Technology in Bratislava, J. Bottu 25, 917 24 Trnava, Slovakia; 2Institute of Materials and Machine Mechanics, Slovak Academy of Sciences, Dúbravská cesta 9, 845 13 Bratislava, Slovakia; 3Department of Experimental Physics, Comenius University in Bratislava, Mlynská dolina F2, 842 48 Bratislava, Slovakia; 4STATON, Ltd., Sadová 1148, 038 53 Turany, Slovakia

**Keywords:** AlTiSiN/TiSiN, nanocomposite coating, PVD, annealing, microstructure, morphology

## Abstract

The present work investigates the influence of isothermal annealing on the microstructure and oxidation behavior of nanocomposite coatings. AlTiSiN/TiSiN coatings with TiSiN adhesive layer were deposited onto a high-speed steel substrate via physical vapor deposition. The coatings were investigated in the as-deposited state as well as after annealing in air at 700, 800, 900 and 1000 °C, respectively. The microstructure and morphology of the coatings were observed using scanning electron microscopy and transmission electron microscopy. The chemical composition and presence of oxidation products were studied by energy-dispersive X-ray spectroscopy. The phase identification was performed by means of X-ray diffraction. In the microstructure of the as-deposited coating, the (Ti_1−x_Al_x_)N particles were embedded in an amorphous Si_3_N_4_ matrix. TiO_2_ and SiO_2_ were found at all annealing temperatures, and Al_2_O_3_ was additionally identified at 1000 °C. It was found that, with increasing annealing temperature, the thickness of the oxide layer increased, and its morphology and chemical composition changed. At 700 and 800 °C, a Ti-Si-rich surface oxide layer was formed. At 900 and 1000 °C, an oxidized part of the coating was observed in addition to the surface oxide layer. Compared to the as-deposited sample, the oxidized samples exhibited considerably worse mechanical properties.

## 1. Introduction

In the machining process, a tool must withstand mechanical and thermal stresses, which can be mitigated by depositing a hard coating layer or multilayer onto its surface. The coating improves the tools utility properties, such as hardness, wear resistance, thermal stability, oxidation resistance and coefficient of friction, thereby improving the quality of the machining process and prolonging the service lifetime [1,2]. The first commercially used physical vapor deposition (PVD) coating for cutting tools was titanium nitride (TiN) [3]. TiN has a high hardness, excellent lubricity and increased wear resistance [3]. A shortcoming of TiN coatings is their poor oxidation resistance at elevated temperatures. When the coatings are exposed to temperatures above 500 °C, they oxidize rapidly, which makes them unsuitable for use at high temperatures [4,5]. The properties of TiN can be enhanced by Al addition [6]. The presence of Al in the TiAlN coating results in an improvement of mechanical properties [7,8,9,10] and increased oxidation resistance [11,12,13]. As such, the TiAlN coatings are some of the most widely used PVD coatings for cutting tools. The utility properties can be further enhanced by introducing different combinations of multilayers with varying chemical compositions, thicknesses and other factors. It was found that the compressive stresses increase significantly with increasing thickness, up to approximately 8 μm, after which they start decreasing [14,15]. Some residual compressive stress in coatings is desirable, since it improves a fatigue lifetime, hinders crack propagation and reduces stress corrosion. Nevertheless, if the compressive stress in the coating is too high, it can reduce the adhesive strength between the coating and the substrate, which may lead to delamination and spalling [14,16].

A nanocomposite coating is a special type of coating. It has excellent physical properties owing to its unique microstructure. The nanocomposite structure can be achieved by, for example, adding Si to the TiN, TiAlN or AlTiN coatings, which results in the formation of amorphous Si_3_N_4_ matrix. The nanocomposite structure consists of nanocrystalline (Ti,Al)N grains embedded in the amorphous matrix. The nanocomposite coatings are suitable for high performance machining. They have favorable wear resistance [17,18] excellent thermal stability and high oxidation resistance [19,20]. The hardness can reach above 40 GPa [21]. The increase in hardness is a result of grain refinement caused by a high density of grain boundaries present in the structure. Since grain boundaries impede the movement of dislocations in the crystalline phase, a plastic deformation is limited. The crystalline phase also blocks a crack propagation in the amorphous phase [22,23,24]. The presence of Si and Al in the coating increases the oxidation resistance due to the formation of protective SiO_2_ and Al_2_O_3_ oxide layers [25]. Concerning the thermal stability, the nanocomposite coatings are stable up to a certain temperature, above which they start losing their beneficial properties due to grain growth. The grain growth causes a shift from nanocomposite to conventional crystalline structure [26]. 

The aim of the present work is to assess the influence of isothermal annealing on the microstructure and properties of AlTiSiN/TiSiN nanocomposite coatings with a TiSiN adhesive layer deposited onto a high-speed steel substrate. During machining, substantial heat is generated at the interaction area of the cutting tool and the workpiece. Therefore, it is essential to study the oxidation behavior at elevated temperatures to avoid an early degradation of the tool [27]. The studies of the high temperature oxidation behavior of AlTiSiN/TiSiN coatings are relatively scarce in scholarly literature despite their significance [25,26]. The oxidation behavior of multicomponent TiAlSiN coatings was studied by Chang and Yang [28]. The authors found that TiAlSiN coatings with a smaller grain size and a higher (Al + Si)/(Ti + Al + Si) content ratio had a lower oxidation rate [28]. The lower oxidation rate was attributed to a protective oxide layer formation, mainly consisting of Al_2_O_3_. The protective scale inhibited the oxygen diffusion into the coating [28]. To the best of our knowledge, the oxidation behavior of TiSiN/AlTiSiN/TiSiN nanocomposite coatings with this combination of layers and substrate has not been investigated yet. To get closer to the real operating conditions of the cutting tool, the studied coatings were annealed in air at high temperatures (700–1000 °C). Subsequently, the morphological changes were investigated based on the examination of the microstructure of the oxidized coatings. 

## 2. Materials and Methods

The AlTiSiN/TiSiN nanocomposite coating was prepared by PVD. The coating consisted of three layers: the TiSiN adhesive layer with the thickness of about 50 nm, the middle AlTiSiN layer (~3 μm) and the top TiSiN layer (~1.5 μm). The thickness of the top TiSiN layer was chosen to be smaller than that of the main AlTiSiN layer to prevent an accumulation of high residual compressive stresses in the coating. The coating was deposited by cathodic arc evaporation onto a high-speed steel substrate (HS6-5-2-5 EN ISO 4957) using PLATIT π80 + DLC with LARC^®^ (lateral rotating cathode) technology. The chemical composition of the substrate is listed in Table 1. 

The substrate had a disc shape with 12 mm diameter and 4 mm thickness. The surface of the substrate was polished and ultrasonically cleaned in acetone prior to deposition. The coating was deposited on one side of the disc. For the coating deposition, pure Al and Ti-Si alloy cathodes were used. For the AlTiSiN layer, the following DC currents were used: 140 A (Al cathode) and 95 A (Ti-Si cathode). For the TiSiN layer, the DC current of 150 A was used (Ti-Si cathode).

The coatings were studied in as-deposited state as well as after annealing at 700, 800, 900 and 1000 °C. The annealing was performed in a laboratory muffle furnace for a period of 1 h in air. The samples were placed directly in the furnace after reaching the annealing temperature. After annealing, the samples were removed and continuously cooled in air. The coating surfaces and cross-sections were observed using JEOL 7600F scanning electron microscope (SEM) operating at the acceleration voltage of 15 kV in regimes of secondary electrons (SEI) or back-scattered electrons (BEI). The chemical compositions of the coatings and oxidation products were studied by energy-dispersive X-ray spectroscopy (EDX) using Oxford Instruments X-max50 spectrometer including INCA software. Microstructural investigations were also performed with a probe-corrected FEI/Thermofisher Scientific Titan Themis 300 transmission electron microscope in scanning (STEM) and conventional (TEM) modes at the accelerating voltage of 200 kV. The STEM was equipped with an EDX system (Super-X). The probe convergence angle was set to 17.5 mrad for imaging applications. 

The phase identification was done by means of X-ray diffraction (XRD) using Panalytical Empyrean X-ray diffractometer with CoKα_1,2_ radiation and PIXCel 3D detector in Bragg–Brentano geometry. A spinning was applied to all samples during the measurement. Measurements were carried out at 40 kV and 40 mA in the angular range between 25° and 90° with the step size of 0.0131°. The counting time of 118 s per pixel per step was used for the detector set to scanning line mode. To verify the presence of an amorphous phase in the coating, the XRD analysis of the as-deposited sample was performed using grazing incidence (GI-XRD) configuration in the angular range between 20° and 100° with the step of 0.02° and 3 s time per step in an open detector mode. Three measurements were made, and their mean was calculated. Furthermore, three measurements of air-scattering were carried out at the same parameters. Subsequently, its effect on the amorphous region in the diffraction pattern was subtracted. The crystal structure of (Ti,Al)N was modeled in VESTA and CrystalMaker software. Initial data were taken from HighScore database (ICSD 98-002-6947 and ICSD 98-060-8628 for TiN and AlN, respectively) and the electron diffraction pattern experimentally observed in this work. For the remaining phases, the following ICSD cards were used: ICSD 98-007-6759 (M_6_C), ICSD 98-061-9056 (MC), ICSD 98-003-6414 (TiO_2_), ICSD 98-016-2612 (SiO_2_), ICSD 01-078-3877 (Al_2_O_3_-R3c) and ICSD 98-017-3713 (Al_2_O_3_-P63/mmc).

Mechanical properties were evaluated by the nanoindentation method using Anton Paar NHT2 device (Berkovich diamond indenter) with a load of 10 mN, loading and unloading rate of 30 mN/min and dwell time of 5 s. For the sample annealed at 800 °C the maximum load of 8 mN was used.

## 3. Results and Discussion

The as-deposited coating was studied by SEM. Figure 1a shows the surface of the coating. A network-like structure and macroparticles can be observed as formed during the deposition process. Figure 1b represents a cross-section of the coating with all the individual layers marked. Layers had a uniform thickness. The EDX elemental map of the cross-section of the sample in the as-deposited state is shown in Figure 1c. The map illustrates that aluminum was only contained in the middle AlTiSiN layer. Moreover, in the TiSiN layers, there were higher contents of titanium, silicon and nitrogen found. 

In Figure 2, the XRD pattern of the coating in as-deposited state is shown. The pattern contains peaks corresponding to the (Ti,Al)N phase. Moreover, the area denoted by the green curve is typical of the amorphous phase, which could represent the amorphous Si_3_N_4_ matrix. The measurement was carried out in such a way that in the resulting XRD pattern the contribution of air-scattering was subtracted (Figure 2). The shapes (FWHM) of the peaks corresponding to the (Ti,Al)N phase suggest that it could consist of small crystals with a large number of grain boundaries, typical of nanoparticles. 

Figure 3 depicts the STEM image of the interface between the AlTiSiN and TiSiN layers. The interface between the layers was uniform. It did not contain any visible irregularities, pores, or voids. The boundary between the AlTiSiN and TiSiN layers was sharp, as is evident from the elemental maps measured at the interface (Figure 3). At the boundary between the layers, fine grains were formed in the direction of layer growth. A detailed view of the individual layers confirms the occurrence of nanocrystals in the amorphous matrix. It can be assumed that nanocrystalline particles are either TiN (in the TiSiN layer mainly) or (Ti,Al)N (in the AlTiSiN layer). Based on the isothermal section of the Al-Ti-N phase diagram [29], an unlimited substitutional solid solution (Ti_1−x_Al_x_)N (x = 0–1) can be formed from the cubic phases TiN and AlN. Both TiN and AlN have a space group no. 225. It is possible that some Ti in TiN was partially substituted by Si, as suggested by Barshilia et al. [30]. However, a phase with a Si:N ratio of 1:1 does not exist in the Si-N system [31]. Only one intermetallic phase, namely Si_3_N_4_, is known in the Si-N system. Therefore, Si was not included in the (Ti_1−x_Al_x_)N designation. In Figure 3, the bottom detail STEM image depicts the nanocrystals in the amorphous matrix. Their size was about 4 nm. According to [3], the (Ti,Al)N particles can be embedded in a minor amorphous Si_3_N_4_ matrix.

Figure 4 shows the selected area electron diffraction pattern (SAEDP) obtained in the region corresponding to the AlTiSiN layer. Based on the results obtained from both XRD and TEM, the (Ti,Al)N phase was identified and its lattice parameter was calculated. Based on the ring radius values measured in SAEDP, it was found that this phase has the lattice parameter of 0.433 nm resulting in an approximate stoichiometry of Ti_0.8_Al_0.2_N. The results are also in agreement with the XRD results. Crystallographic data and structural parameters used for modeling of the (Ti,Al)N phase are shown in Table 2. The distribution of Ti and Al atoms in the modeled crystal structure of (Ti,Al)N corresponds approximately to the Ti:Al atomic ratio of 4:1.

The surface morphologies of the coatings annealed at different temperatures are shown in Figure 5. At 700 °C (Figure 5a), no significant difference in the surface morphology compared to the as-deposited state was observed. Similarly, at 800 °C (Figure 5b), the sample surface was not changed much. However, the network-like structure was no longer visible, possibly due to a formation of thin oxide layer on the surface at this temperature. More substantial changes in the surface morphology were observed at 900 °C (Figure 5c). At 900 °C, the whole surface was covered with a continuous oxide scale. The scale had a blade-like and needle-like morphology. Finally, at the highest temperature of 1000 °C, the blade-like structures have grown substantially, forming an oxide scale that uniformly covered the entire surface (Figure 5d).

EDX elemental maps of the cross-sections of the annealed samples are given in Figure 6. The corresponding chemical composition is summarized in Table 3. The amount of oxygen increases with increasing annealing temperature. At the same time, the amount of nitrogen decreases. The ratio between Ti, Si and Al remains stable upon all annealing temperatures. After annealing at 700 and 800 °C, oxygen was detected on the surface in the form of a thin oxide layer corresponding to either titanium and/or silicon oxides. At 800 °C, the oxide layer has grown slightly thicker compared to the one observed at 700 °C. After annealing at 900 °C, a more pronounced oxidation of the coating occurred. A Ti-Si-rich oxide surface layer was formed on the top of the coating. Furthermore, an oxidation of the part of the TiSiN layer also occurred. This oxidized layer had an increased Si concentration. Finally, at 1000 °C the oxide/oxidized layer grew thicker not only in the outwards direction, but also inwards, consuming more than half of the original coating thickness. A similar phenomenon was observed by Parlinska-Wojtan [32]. As follows from Figure 6, the outer oxide layer with the blade-like morphology was rich in Ti and, interestingly, Al, some of which diffused there from the AlTiSiN layer. On the contrary, Si was not detected in the outer oxide layer but in the oxidized layers only, along with Ti. At 1000 °C, the entire TiSiN layer and part of the AlTiSiN layer were oxidized.

The XRD patterns of the as-deposited and as-oxidized coatings are given in Figure 7. In the XRD pattern of the as-deposited coating, peaks (111) and (002) corresponding to cubic (Ti,Al)N were identified (Figure 7a). The peaks were also found in previous studies [21,30,33,34,35,36,37,38,39,40,41], sometimes with slightly shifted 2θ values. This shift is caused by substitution of Ti by Al, and possibly Si, and also due to the effects of crystallite size and internal stresses [30]. Additionally, peaks of ferrite and carbides MC and M_6_C, corresponding to the substrate, were identified in the XRD pattern (Figure 7a).

As follows from SEM/EDX observations (Figure 6), the isothermal annealing of the coating at different temperatures resulted in the formation of an oxide/oxidized layer on the surface. It was observed that, with increasing annealing temperature, the oxide/oxidized layer grew thicker, and its morphology was changed. The oxidation rate was faster at higher temperature due to the faster diffusion of reactants. Figure 8 shows the sequence of oxides at 700–1000 °C. At 700, 800 and 900 °C, oxide layers rich in Ti and Si were formed on the surface. TiO_2_ and SiO_2_ were identified in the XRD patterns at 800 °C and 900 °C, respectively. At 700 °C, only TiO_2_ was found. The lack of SiO_2_ peaks in the XRD pattern is attributable to its small thickness. The presence of SiO_2_ was confirmed by EDX (Figure 6); however, this oxide was probably present in a small quantity. As such, it was not detectable by XRD. At 1000 °C, the EDX analysis (Figure 6) showed a presence of Al in the top oxide layer. This observation was backed up by the XRD analysis (Figure 7e) showing several Al_2_O_3_ peaks. The presence of aluminum oxide is surprising as it was not observed at lower temperatures. The formation of Al_2_O_3_ was observed during the oxidation of the AlTiSiN layer. It was not observed at lower temperatures, where only the TiSiN layer was oxidized. It is possible that some Al diffused outwards from the AlTiSiN layer. The diffusion of Al was studied in papers by Parlinska-Wojtan [32] and Vennemann et al. [42]. In the former study, the formation of Al_2_O_3_ on the coating surface was explained by a high diffusion rate of Al in Ti_1−x_Al_x_N coatings at high temperatures. However, other studies [43,44] revealed that the diffusivity of Al in similar systems is comparable to or lower than the diffusivity of Ti. The TiSiN layer is a sufficient diffusion barrier for Al [45]. Therefore, the formation of Al_2_O_3_ was only possible when the TiSiN layer was fully oxidized.

A further insight into the Al behavior could be obtained by investigating the Gibbs energies of particular oxides. Ti, Si and Al have a different affinities towards oxygen [46]. Figure 9 shows the Gibbs energies of Al_2_O_3_, TiO_2_ and SiO_2_. Data in Figure 9 are taken from studies by Zheng et al. [47] and Li et al. [48]. The stability of the oxides increases in the following manner:SiO_2_ ˂ TiO_2_ ˂ Al_2_O_3_(1)

Aluminum oxide is the most stable. Therefore, Al could have diffused to the top oxide layer where it reacted with oxygen. Furthermore, the following reaction is thermodynamically possible:4Al + 3SiO_2_ → 3Si + 2Al_2_O_3_(2)

This reaction shows that thickness of SiO_2_ is reduced in the presence of Al. The driving force for this reaction is given by the difference between the Gibbs energies of SiO_2_ and Al_2_O_3_, which is approximately 170 kJ mol^−1^ (Figure 9). This difference is high enough to prevent the oxidation of Si in the presence of Al. Therefore, SiO_2_ was not observed in the top oxide layer.

The TiSiN layer was predominantly oxidized at 700–900 °C. The Gibbs energy of TiO_2_ is smaller than SiO_2_; however, the difference is less than 20 kJ mol^−1^ (Figure 9). Such a small difference is insufficient to prevent the oxidation of Si in the presence of Ti. Therefore, both SiO_2_ and TiO_2_ were present in the oxidized TiSiN layer. The relative nobility of Si compared to Ti was, however, reflected in a smaller amount of SiO_2_ formed compared to TiO_2_. At 700 °C, this presumption was also confirmed experimentally by XRD (Figure 7b).

Ti and Al have a greater affinity towards oxygen compared to Si (Figure 9). Therefore, both TiO_2_ and Al_2_O_3_ were observed in the top oxide layer at 1000 °C. Al has a higher affinity towards oxygen compared to Ti. Therefore, the following reaction is possible
4Al + 3TiO_2_ → 3Ti + 2Al_2_O_3_(3)

The top oxide scale at 1000 °C grows by outward diffusion and oxidation of Al at the expense of TiO_2_. This is reflected by the presence of Al_2_O_3_ in the voids of TiO_2_ (Figure 6). The difference in Gibbs energies of TiO_2_ and Al_2_O_3_ is approximately 150 kJ mol^−1^ (Figure 9). This difference is probably not high enough to completely avoid the TiO_2_ formation. An oxidation of TiAlSiN coatings was previously studied by Bak and Lee [49]. Their TiAlSiN coatings initially oxidized to an Al_2_O_3_ layer with some TiO_2_, below which a rutile-TiO_2_ layer containing Al_2_O_3_ was formed [49]. At longer annealing times, another TiO_2_-rich surface layer was formed over the previously formed (Al_2_O_3_-rich)/(TiO_2_-rich) bilayers, as Ti continuously diffused outwards from the film. The authors had Ti-excess TiAlSiN films (Ti_0.26_Al_0.163_Si_0.012_N_0.565_, [49]). Therefore, the reaction (3) was partially reversed towards TiO_2_ formation because of the high Ti concentration.

A slight mismatch between the AlTiSiN layer (including the oxidized part of the layer) in the sample annealed at 1000 °C has been observed (Figure 8). It might have two possible explanations. It could have happened due to the inward oxidation and Al diffusion towards the surface, causing the region in the AlTiSiN layer to be depleted of Al, which resulted in slightly reduced thickness of the layer. The other possibility is that in this coating, the AlTiSiN layer was deposited to a slightly smaller thickness, which was caused by the directional character of the PVD processes. 

More information about the degradation behavior of the studied coatings could be obtained by investigating their mechanical properties. Nanohardness and elastic modulus were studied. A summary of mechanical properties is listed in Table 4. Furthermore, the results are shown graphically in Figure 10. Only results for the as-deposited coating and coatings annealed at 700 and 800 °C are included. The coatings annealed at 900 and 1000 °C were significantly degraded by oxidation. It was not possible to measure their hardness accurately due to delamination.

The hardness of the as-deposited coating was 35.41 GPa. The hardness value of the as-deposited coating is comparable to previous studies investigating AlTiSiN coatings. For example, Xiao et al. [21] measured the hardness of 36.1 GPa for the AlTiSiN coating, Chang and Cai [13] found a slightly higher value of 38 GPa, and Tillmann and Dildrop [50] obtained 33 GPa. The samples annealed at 700 and 800 °C showed hardnesses of 20.74 GPa and 18.86 GPa, respectively. As expected, the annealed samples exhibited much lower hardness due to degradation of the coating caused by oxidation. The elastic moduli of the coatings in the as-deposited state and after annealing at 700 and 800 °C were 350.62 GPa, 371.49 GPa and 324.99 GPa, respectively. These values did not correlate well with the elastic moduli values from abovementioned studies [13,21,50] due to differences in chemical composition. A similar phenomenon was also noted by Musil et al. [51,52].

The H/E ratio is related to the elastic deformation [51]. The H^3^/E^2^ ratio expresses the resistance to plastic deformation [51]. The as-deposited coating showed the highest value of H/E (0.101), while the coatings oxidized at 700 and 800 °C exhibited lower values of 0.056 and 0.058, respectively. Similarly, the highest H^3^/E^2^ ratio was found for the as-deposited sample (0.36). The two coatings oxidized at 700 and 800 °C had much lower values of 0.065 and 0.063, respectively. It means that the as-deposited coating exhibited a much higher resistance to plastic deformation compared to the annealed samples. According to Chen et al. [53], the coatings with H/E > 0.1 do not form cracks, while those with H/E < 0.1 do. Considering this assumption, the samples annealed at 700 and 800 °C with H/E ratios of 0.056 and 0.058, respectively, can form cracks more easily during an external load compared to as-deposited coating. Chen et al. [53] also state that a coating with a larger H^3^/E^2^ is less likely to be plastically deformed and is therefore tougher. Coming from this assumption, the as-deposited coating shows a higher resistance to plastic deformation (higher toughness) compared to the annealed coatings. The above results demonstrate that the oxidized samples had substantially worse mechanical properties.

## 4. Conclusions

The present work contributes to a fundamental understanding of the influence of isothermal annealing on the resulting microstructure and morphology of the AlTiSiN/TiSiN nanocomposite coatings. The obtained results can be summarized as follows:The AlTiSiN/TiSiN nanocomposite coatings were prepared by PVD. The coatings consisted of three layers: the TiSiN adhesive layer with the thickness of about 50 nm, the middle AlTiSiN layer (∼3 μm) and the top TiSiN layer (~1.5 μm).The transition from the AlTiSiN layer to the TiSiN layer had a sharp boundary. At the boundary between the layers, fine grains were formed in the direction of layer growth. A detailed view of the individual layers confirmed the presence of nanocrystals in the amorphous matrix. It can be assumed that nanocrystalline particles were either TiN (in the TiSiN layer mainly) or (Ti,Al)N (in the AlTiSiN layer).Based on the experimental results from the TEM and XRD analyses, crystallographic data and structural parameters were obtained for the Ti_0.8_Al_0.2_N compound identified in the as-deposited coating.The coatings were oxidized at 700–1000 °C in air. TiO_2_ and SiO_2_ were found at all annealing temperatures. At 1000 °C, Al_2_O_3_ was additionally identified. A significant change in surface morphology was observed in the samples annealed at 900 and 1000 °C, where the whole surface was covered with a thick oxide layer with blade-like and needle-like morphology. A network-like structure, on the contrary, was observed at the lower annealing temperature (700 °C), similar to the as-deposited sample.At 700 and 800 °C, the Ti-Si-rich surface oxide layer was formed. At 900 and 1000 °C, an oxidized part of the coating was observed in addition to the surface oxide layer. It was found that Al diffused from the AlTiSiN layer to the surface of the coating, where it reacted with oxygen and formed Al_2_O_3_.The samples oxidized at 700 and 800 °C exhibited considerably worse mechanical properties compared to the as-deposited sample. The samples oxidized at 900 and 1000 °C were significantly degraded by oxidation. As such, it was not possible to measure their nanohardness accurately.

## Figures and Tables

**Figure 1 nanomaterials-13-00474-f001:**
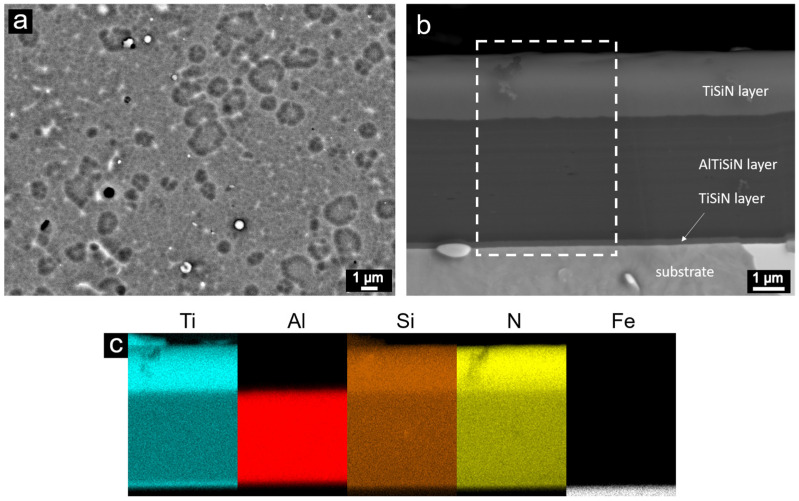
Surface (**a**), cross-section (the white dashed line rectangle marks the area where the EDX mapping was performed) (**b**) and elemental maps of cross-section (**c**) of the coating in as-deposited state, as documented by SEM/EDX. Cyan, red, brown, yellow and white colors correspond to Ti, Al, Si, N and Fe, respectively.

**Figure 2 nanomaterials-13-00474-f002:**
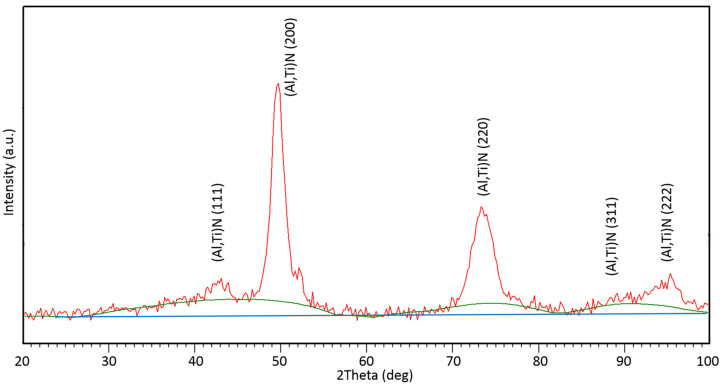
XRD pattern of the coating in as-deposited state (red curve); blue line corresponds to background, and green curve represents the area of the amorphous phase.

**Figure 3 nanomaterials-13-00474-f003:**
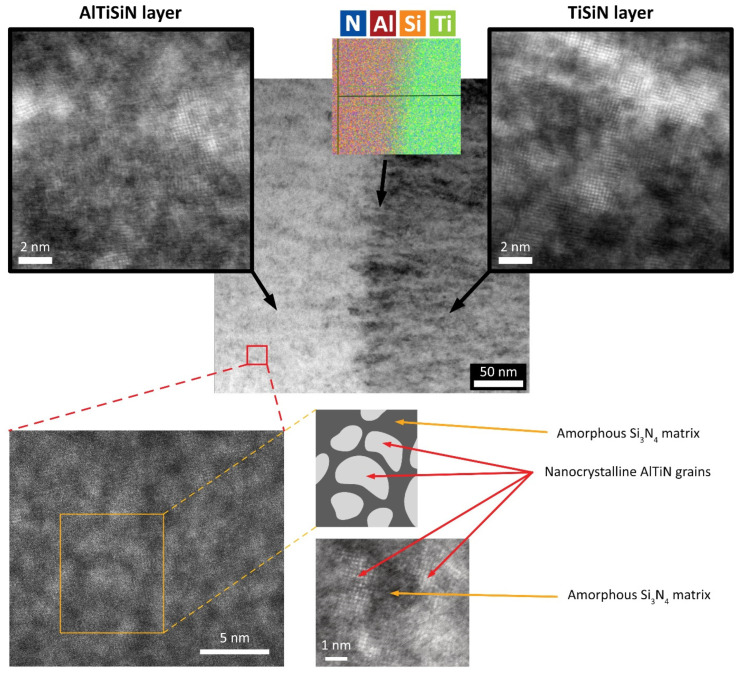
STEM image of the AlTiSiN/TiSiN layer interface including EDX elemental maps. The bottom images show the detailed view and a schematic of the nanocomposite structure.

**Figure 4 nanomaterials-13-00474-f004:**
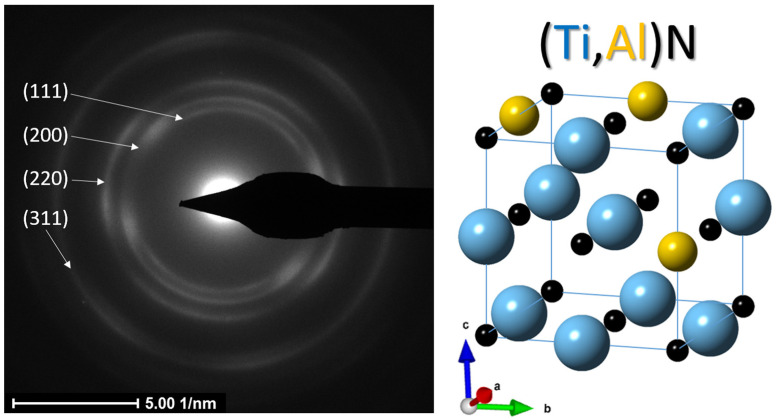
Electron diffraction pattern of (Ti,Al)N and corresponding modeled crystal structure.

**Figure 5 nanomaterials-13-00474-f005:**
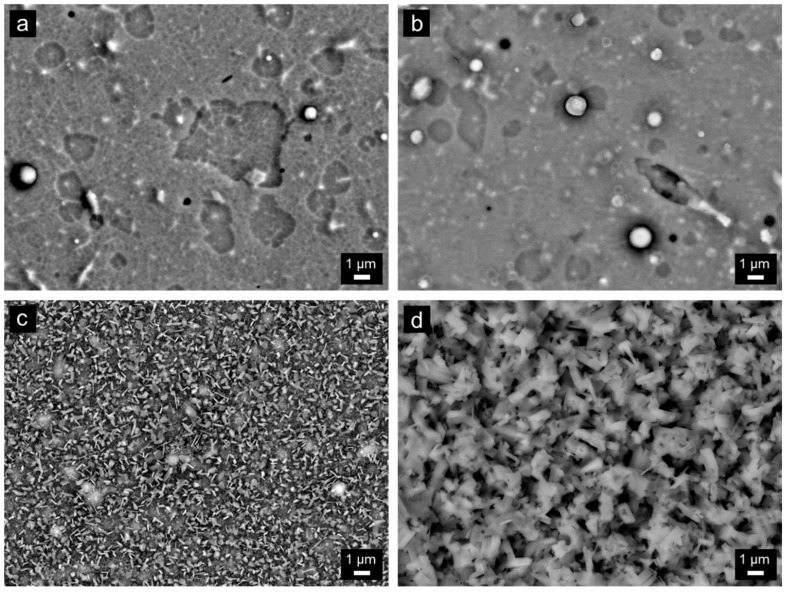
SEM images of the surface morphology of the coatings after annealing in air at 700 °C (**a**), 800 °C (**b**), 900 °C (**c**) and 1000 °C (**d**) for 1 h.

**Figure 6 nanomaterials-13-00474-f006:**
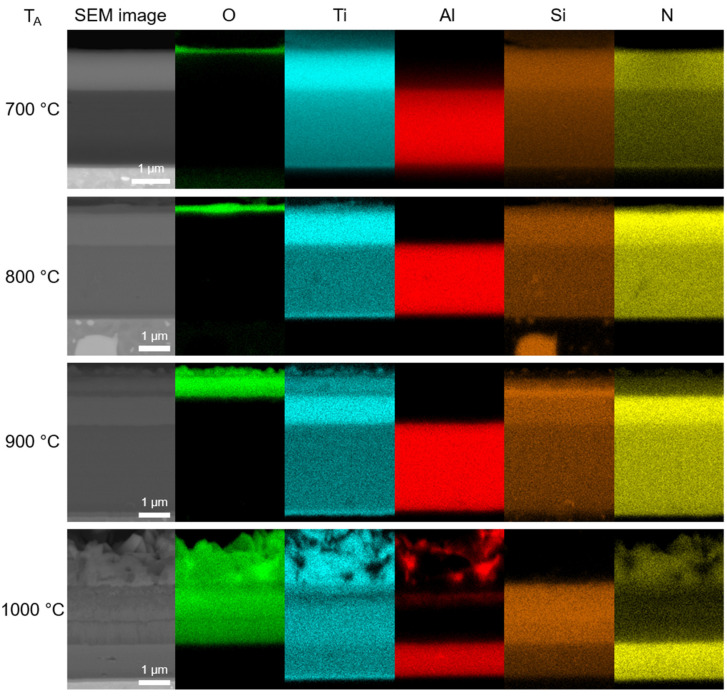
EDX elemental maps of the cross-section of annealed coatings (T_A_—annealing temperature).

**Figure 7 nanomaterials-13-00474-f007:**
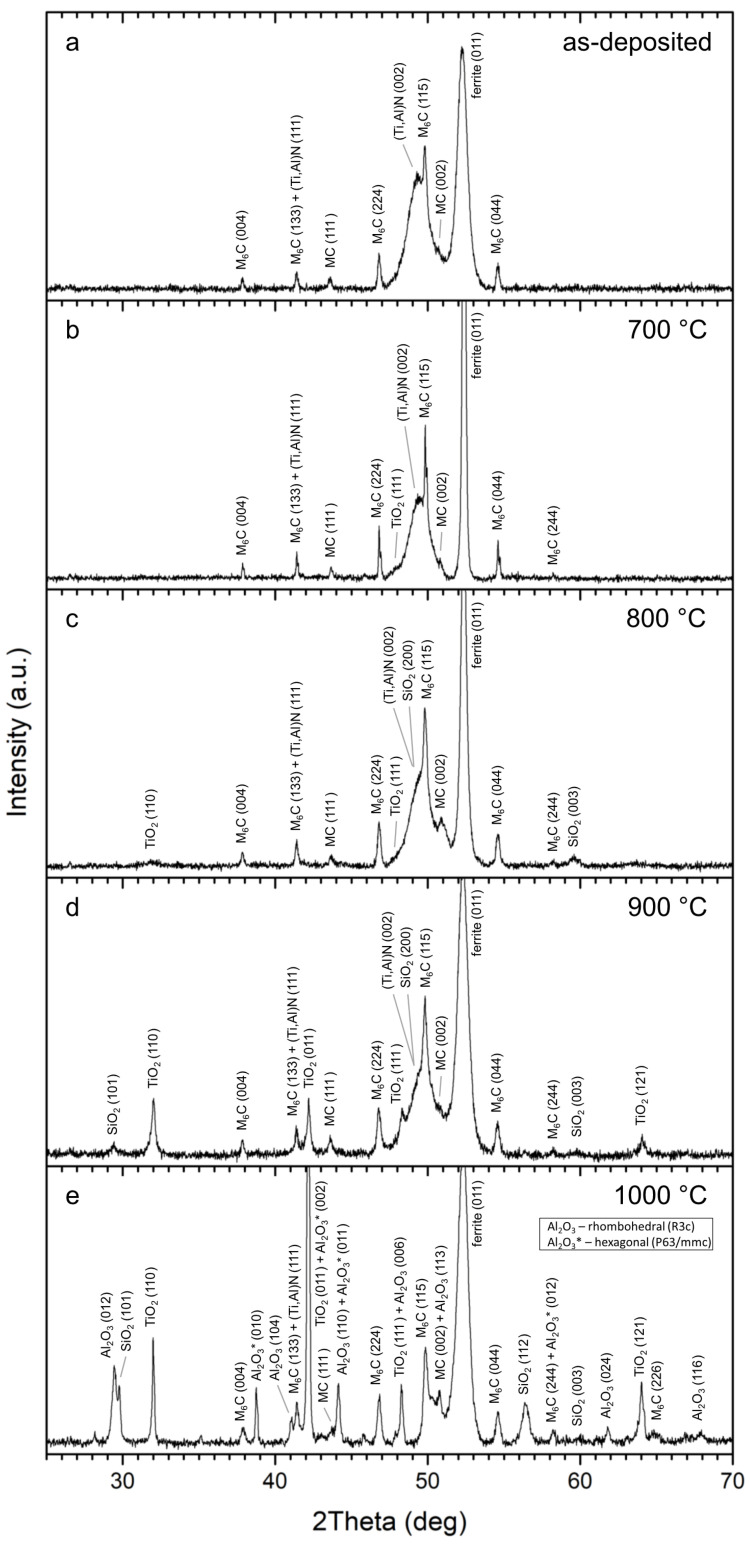
XRD patterns of the investigated samples: as-deposited state (**a**) and annealed at 700 °C (**b**), 800 °C (**c**), 900 °C (**d**) and 1000 °C (**e**).

**Figure 8 nanomaterials-13-00474-f008:**
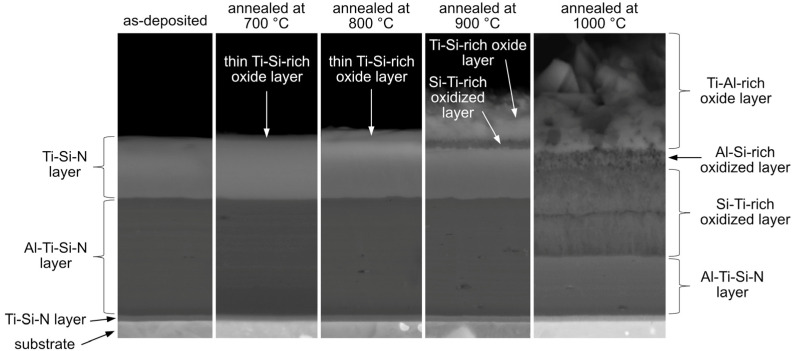
Overview of the layers formed in the investigated coatings.

**Figure 9 nanomaterials-13-00474-f009:**
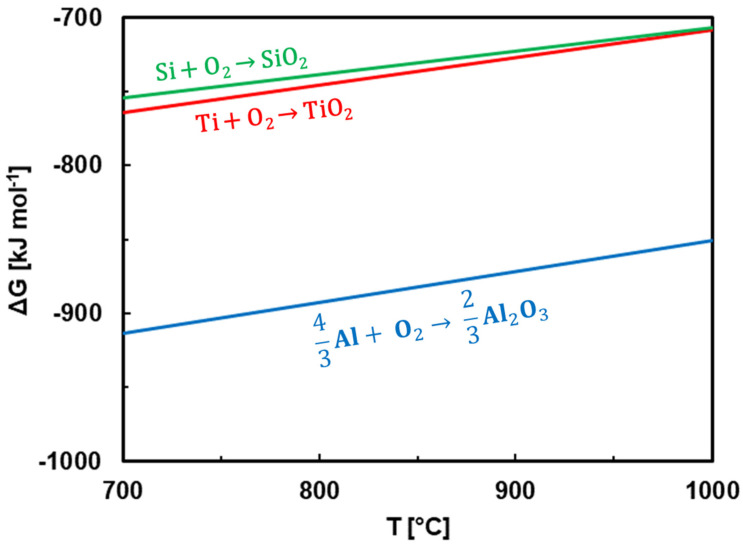
Gibbs free energies of formation of SiO_2_, TiO_2_ and Al_2_O_3_.

**Figure 10 nanomaterials-13-00474-f010:**
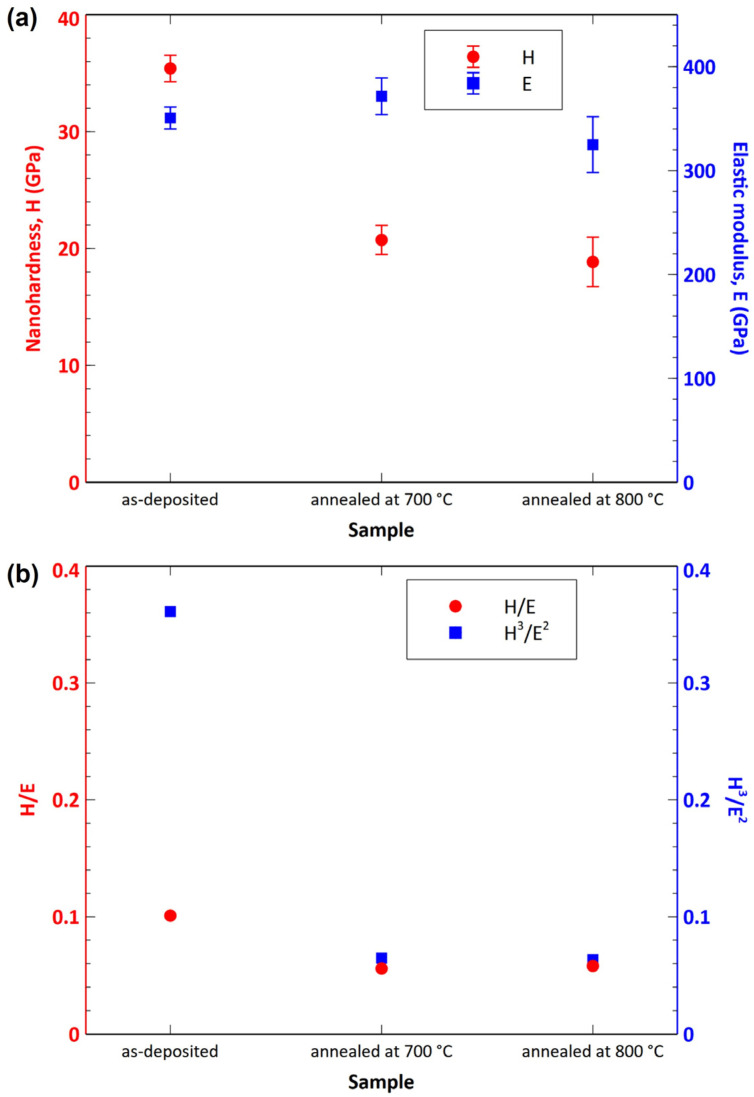
Hardness, elastic modulus (**a**) and the H/E and H^3^/E^2^ ratios (**b**) of the studied samples.

**Table 1 nanomaterials-13-00474-t001:** Chemical composition of the substrate HS6-5-2-5 EN ISO 4957.

Chemical element	C	Mn	Si	Cr	W
**Content (wt. %)**	0.87–0.95	max 0.40	max 0.45	3.80–4.50	5.90–6.70
**Chemical element**	**V**	**Mo**	**Co**	**P**	**S**
**Content (wt. %)**	1.70–2.10	4.70–5.20	4.50–5.00	max 0.03	max 0.03

**Table 2 nanomaterials-13-00474-t002:** Crystallographic data and structural parameters of (Ti,Al)N.

Crystallographic Data of (Ti,Al)N				
Pearson Symbol	Space Group	Symmetry	a_TEM_ [nm]	
cF8	225	$Fm\bar{3}m$	0.433	
**Structural Parameters of (Ti,Al)N**				
**Atom**	**x**	**y**	**z**	**Site Occupancy**
Ti	0.5	0.5	0.5	0.8
Al	0.5	0.5	0.5	0.2
N	0	0	0	1

**Table 3 nanomaterials-13-00474-t003:** Chemical composition of AlTiSiN/TiSiN coatings measured by EDX in the cross-sections.

Sample Condition	Chemical Composition [at. %]				
	N	O	Al	Si	Ti
as-deposited	55.9	-	13.9	7.5	22.7
700 °C	52.1	6.2	13.8	7.7	22.2
800 °C	47.4	9.8	13.5	7.9	21.4
900 °C	38.7	22.4	12.4	7.2	19.3
1000 °C	7.8	62.4	8.9	5.9	15.0

**Table 4 nanomaterials-13-00474-t004:** Nanohardness (H), elastic modulus (E) and the calculated H/E and H^3^/E^2^ ratios of the studied samples.

	As-Deposited	Annealed at 700 °C	Annealed at 800 °C
**H [GPa]**	35.41	20.74	18.86
**E [GPa]**	350.62	371.49	324.99
**H/E**	0.101	0.056	0.058
**H^3^/E^2^**	0.361	0.065	0.063

## Data Availability

Data are available from the corresponding author upon request.

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
