# Peer review of "Influence of Isothermal Annealing on Microstructure, Morphology and Oxidation Behavior of AlTiSiN/TiSiN Nanocomposite Coatings"

_nanomaterials, 2023, doi:10.3390/nano13030474_

Round 1
Reviewer 1 Report
Dear authors,
Your attempt on investigating the influence of isothermal annealing on the microstructure and oxidation behavior of nanocomposite AlTiSiN/TiSiN coatings with TiSiN adhesive layer via physical vapor deposition, is of great interest for scientific community. The spectroscopic and other techniques used, were previously shown to be efficient on characterizing such a process. However, I would suggest minor revisions, in order to improve some minor mistakes exist in your manuscript, which are included in the attached pdf file.

Reviewer 2 Report
The article presents a study regarding the influence of the isothermal annealing on microstructure, morphology, oxidation behavior and mechanical properties of AlTiSiN/TiSiN nanocomposite coatings. The authors must clarify some aspects before publication (minor revision). Specific comments are given bellow:
1. In “Introduction” section, the authors must emphasize better the novelty of their study in comparison to the data reported relatively recently in literature (in the last 5 years) regarding the influence of thermal annealing on mechanical properties of AlTiSiN/TiSiN coatings.
2. In “Materials” section, the authors must provide more experimental details for the “isothermal annealing” process (heating rate, cooling rate, etc.)
3. The authors must provide ICDD card no. for phase identification in the XRD patterns.
4. Besides EDX maps, the authors must add tables containing the atomic percentage of elements for AlTiSiN/TiSiN nanocomposite coatings (before and after annealing).
5. In the SAED pattern image, the authors must add the association between the diffraction rings and compound phase.
